# LAuReL: Learned Augmented Residual Layer

**Gaurav Menghani** [1]  **Ravi Kumar** [1]  **Sanjiv Kumar** [2]

## Abstract

One of the core pillars of efficient deep learning methods are architectural improvements, such as residual/skip connections, which have led to significantly better model convergence and quality. Since their introduction, residual connections have become ubiquitous not only in convolutional neural networks but also in transformer-based architectures, the backbone of LLMs.

In this paper, we introduce the *Learned Augmented Residual Layer* (LAuReL)—a novel generalization of the canonical residual connection—designed to serve as an in-situ replacement while outperforming it in both model quality and footprint metrics. Our experiments show that LAuReL can enhance quality for both vision and language models while adding fewer parameters and incurring less latency and memory overhead than naively increasing parameter count.

For example, on the ImageNet-1K task, LAuReL achieves the same model quality improvements as naively adding an extra layer while using $2.6\times$ fewer parameters. Similarly, when pre-training 1B and 4B parameter LLMs, LAuReL improves performance on a variety of challenging downstream evaluation tasks by 2.54% to 20.05%, while adding only 0.012% and 0.1% additional parameters, respectively.

## 1. Introduction

Model efficiency is of critical importance in the age of extremely large language and vision models. Even if a given model has impressive quality, its footprint metrics such as train-time compute, inference latency, resident memory size, peak memory consumption, etc. dictate if it can be experimented with and/or deployed in real-world settings. A large and slow model can be impractical to train and use, making it unsuitable for applications that require fast responses, no matter how well it performs on benchmarks.

LLMs such as Gemini 1.5 Flash (Gemini-Team et al., 2024), DeepSeek V3 (DeepSeek-A.I. et al., 2024) have been explicitly designed with these efficiencies in mind and consistently outperform much larger models that preceded them. Consequently, improving the Pareto-frontier of model quality and model footprint, via efficient learning methods has been an area of active research in the past few years. Areas of interests span from algorithmic techniques (Menghani, 2023) and efficient hardware (Sze et al., 2017) to best practices around model efficiency (Dehghani et al., 2022).

One of the core pillars of efficient deep learning methods are architectural improvements such as the residual/skip connection, which had led to significantly better model convergence and quality (He et al.). The residual connection has become ubiquitous not only in convolutional neural networks but also in transformer-based architectures (Vaswani et al., 2017), which are the backbone of today's LLMs.

In this paper we introduce *learned augmented residual layer*, LAuReL, which generalizes the canonical residual connection. Recall that deep-learning models with residual connections have a 'block' structure, with many blocks chained together between the input and final output; these could be convolution/identity blocks within a ResNet, a transformer block in a transformer encoder/decoder, etc. Within a block, a typical residual connection is given by:

$$x_{i+1} = f(x_i) + x_i. \qquad (1)$$

Here, $f(\cdot)$ can be any non-linear function such as attention, MLP, multiple non-linear layers, etc., $x_i$ is the input to the said non-linear function, and $x_{i+1}$ is the combined output of the non-linear function and the residual component (Figure 1). For simplicity, we ignore pre-processing functions such as layer norm, which can be folded into $f(\cdot)$.

## 2. Learned Augmented Residual Layer

In this section we describe the main idea behind LAuReL. In its most general form, we reformulate the residual con-

---

[1]Google Research, Mountain View, CA. gmenghani@google.com, ravi.k53@gmail.com [2]Google Research, New York, NY. sanjivk@google.com. Correspondence to: Gaurav Menghani <gmenghani@google.com>.

*Proceedings of the $42^{nd}$ International Conference on Machine Learning*, Vancouver, Canada. PMLR 267, 2025. Copyright 2025 by the author(s).

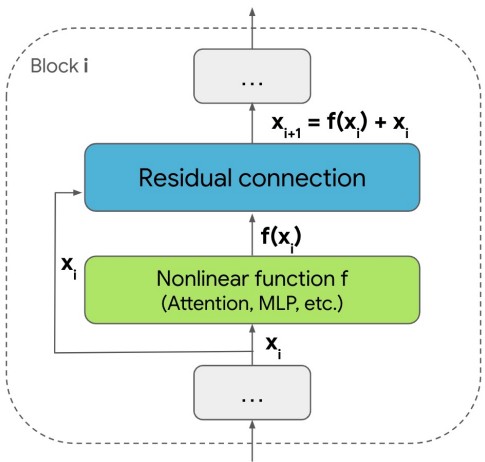

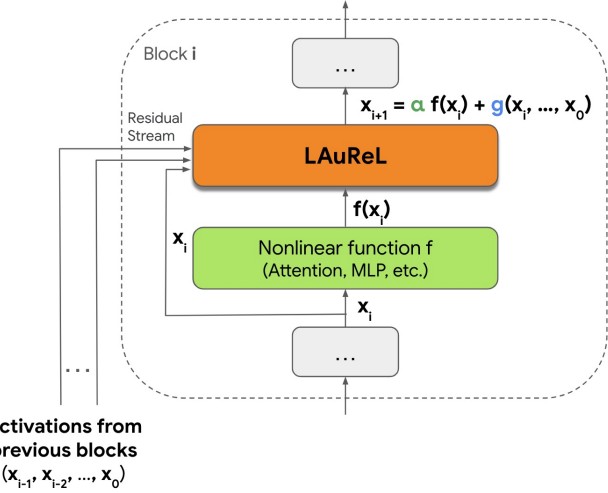

*Figure 1.* A standard residual connection. We assume the model to be divided into logical 'blocks', which is true for most modern architectures including transformers. The residual connection combines the output of a non-linear function $f$ and the input to the said non-linear function. Here, $f$ can be attention, MLP, or any other combination of non-linear layers.

*Figure 2.* An illustration of the LAUREL framework; see (2). LAUREL can be used to replace the regular residual connection in Figure 1. Again, $f$ can be any non-linear function such as attention, MLPs, and groups of multiple non-linear layers.

nection to be the following:

$$x_{i+1} = \alpha \cdot f(x_i) + g(x_i, x_{i-1}, \ldots, x_0). \quad (2)$$

Here, $\alpha$ is a learned scalar parameter, and $g(\cdot)$ is a learned linear function with $x_i, x_{i-1}, \ldots, x_0$ as inputs, where $x_j$ is the output of the $j$-th residual connection.

The main intuition is that one can learn a richer set of (linear) functions than just using $x_i$ as the residual component. One motivation behind seeking these richer linear functions is the concept of a "residual stream" (Elhage et al., 2021), where the residual connection is considered to be part of a stream of information that passes through each layer without being exposed to any non-linearities. This allows the learning process to focus on the non-linear components better.

Each layer/operation can read from, and subsequently write to this residual stream. Since the residual connection has been shown to be important for model quality and convergence, we designed LAUREL to operate on this residual stream in a learned fashion, while being light-weight in terms of the model size and latency changes.

In this paper we study three specific versions of the LAUREL framework; although as described in (2), the framework can be generalized beyond these versions.

### 2.1. Residual Weights Version (LAUREL-RW)

In this version, we keep $\alpha$ learnable and set $g(x_i, \ldots, x_0) = \beta x_i$. Therefore, (2) can be rewritten as:

$$x_{i+1} = \alpha f(x_i) + \beta x_i.$$

Notice that this version assigns learnable weights to the $f(x_i)$ and $x_i$ from (1). In practice, we found that we cannot let $\alpha$ and $\beta$ grow unbounded, and using a normalization function such as `softmax` or `sigmoid` helps. Clearly, this version will add only two new parameters per LAUREL layer. If necessary, we can replace these two parameters by a single learnable parameter and use a function such as `sigmoid` to define $\alpha, \beta$ in terms of this single parameter.

This variant can be useful for learning the relative importance of the non-linear component ($f(x_i)$) and the residual input ($x_i$). In the earlier layers the former might be more important, while in the later layers, the latter could be useful for mitigating problems such as vanishing gradients. This variant can help by adaptively learning these weights.

### 2.2. Low-Rank Version (LAUREL-LR)

Here, we fix $\alpha = 1$, and $g(x_i) = Wx_i$ in (2) to obtain:

$$x_{i+1} = f(x_i) + Wx_i.$$

As written, $W$ is a learnable $D \times D$ matrix, where $D$ is the model dimension for transformer-based models, or more generally it is the last dimension of $x_i$. Hence, the $W$ matrix will add $D^2$ new parameters (per LAUREL layer).

In practice, to reduce the number of new parameters added to the model and to help with convergence, we consider a low rank version of $W$. In particular, let $W = A \times B + I$, where $A$ and $B^T$ are $D \times r$ matrices and $r \ll D$. Thus, we can rewrite (2) as:

$$x_{i+1} = f(x_i) + BAx_i + x_i. \quad (3)$$

Here, both $A$ and $B$ matrices are learnable. The number of new parameters is $2rD$, per LAUREL layer.

This variant helps with allocating learning capacity for the linear part of the network (the residual input, i.e., $x_i$ in Figure 2), such that the main network can use its capacity towards learning better nonlinear functions ($f(x_i)$), while LAUREL contributes the linear components ($x_i + BAx_i$) to the residual stream.

### 2.3. Previous Activations Version (LAUREL-PA)

This is similar to LAUREL-LR, except that we use $k$ activations from the previous blocks. In particular, we set

$$g(x_i, \ldots, x_0) = x_i + \sum_{j=0}^{k-1} \gamma_{i,j} \cdot h_i(x_{i-j}),$$

where $\gamma_{i,0}, \ldots, \gamma_{i,k-1}$ are learned scalar parameters and $h_i$ is another linear function.[1] This allows us to rewrite (2) as:

$$x_{i+1} = f(x_i) + x_i + \sum_{j=0}^{k-1} \gamma_{i,j} \cdot h_i(x_{i-j}). \qquad (4)$$

In practice, we replace $h_i$ by a low-rank product similar to the LAUREL-LR version, but using the identity function is also an option. When using a rank $r$ product for $h_i$, the number of new parameters per LAUREL layer is $2rD + k$, where $k$ is the number of previous activations used.

We can consider this variant to be a hybrid of the LAUREL-RW and LAUREL-LR variants, where multiple previous activations are used in a weighted manner that is learned. This allows layers accelerated access to previous activations, along with learning their relative importance.

### 2.4. Other Derived Variants

All three proposed LAUREL versions are a combination of scalar and/or low-rank products on top of the vanilla residual connection in (1). This makes it especially light-weight in terms of its impact on model size and latency. We discuss the efficiency of the variants in more detail in Section 4.

That being said, LAUREL is generic enough to allow combinations of the above variants, as well as new variants. For instance, one straight-forward combination is LAUREL-RW+LR, where the residual weights from LAUREL-RW can be applied along with the LAUREL-LR variant to rewrite (3) as follows:

$$x_{i+1} = \alpha f(x_i) + \beta(BAx_i + x_i). \qquad (5)$$

Similarly, LAUREL-PA as defined in (4) can be combined

---

[1]For simplicity, we fix $\alpha = 1$.

with LAUREL-RW as follows:

$$x_{i+1} = \alpha f(x_i) + \beta \left( x_i + \sum_{j=0}^{k-1} \gamma_{i,j} \cdot h_i(x_{i-j}) \right).$$

When using a low-rank product for $h_i$, we can create the LAUREL-RW+LR+PA variant as follows:

$$x_{i+1} = \alpha f(x_i) + \beta \left( x_i + \sum_{j=0}^{k-1} \gamma_{i,j} \cdot A_{i,j} B_{i,j}(x_{i-j}) \right). \tag{6}$$

Yet another variant while slightly relaxing our above formulations would be to treat $\alpha$ and $\beta$ as vectors of length $D$. Here, we would learn fine-grained per-dimension weights when mixing $x_i$ and $f(x_i)$ at the cost of $2D$ parameters per LAUREL-LR layer, instead of two per parameters per layer.

To summarize, LAUREL is inherently flexible and provides many possible cheap learnable augmentations on top of the vanilla residual connection. In the following section we demonstrate that these combinations are efficient and effective at improving the model quality of common vision and language models, while having a minimal impact on the number of parameters and latency.

## 3. Experiments

We experiment with LAUREL in two domains, namely, vision and language. For the first case, our goal is to improve the image classification accuracy of the ResNet-50 model on the ImageNet-1K dataset (Deng et al., 2009). For the second case our goal is to improve the performance of two different large language models (LLMs) of size 1B and 4B parameters respectively, evaluated after the pre-training stage, on common benchmarks.

The underlying motivation behind these experiments is not necessarily to improve on the SOTA results, but to show how LAUREL can be easily integrated on top of common model architectures with residual/skip connections in order to achieve a better model quality and footprint trade off.

### 3.1. ResNet-50 on ImageNet-1K

In this setup we train a standard ResNet-50 model on the ImageNet 1K dataset (Deng et al., 2009) using 16 Google Cloud TPUv5e chips over one epoch with data-augmentation turned on. In order to obtain a strong baseline, we fine-tuned the model learning rate schedule and picked a schedule that maximized the average of the best accuracy@1 values over 5 trials (which we simply refer to as accuracy in this subsection). The baseline model that we obtained achieves an accuracy of $74.95 \pm 0.016\%$.

In addition, we also find that if we simply add another layer to the ResNet-50 model (i.e., naive scaling), we can

*Table 1.* Applying LAUREL on a ResNet-50 trained on the ImageNet 1K classification dataset. LAUREL-RW provides a significant boost with negligible extra parameters. LAUREL-RW+LR and LAUREL-RW+LR+PA meet and beat the naive scaling baseline while using $2.6\times$ and $1.82\times$ fewer parameters. Results that provide statistically significant boost over the baseline are highlighted in green and bold.

| MODEL | AVG. BEST ACCURACY@1 (%), 5 TRIALS | PARAMS ADDED VS BASELINE (%) |
|---|---|---|
| BASELINE | $74.95 \pm 0.01$ | 0.00 |
| BASELINE + 1 LAYER (NAIVE SCALING) | $75.20 \pm 0.12$ | 4.37 |
| LAUREL-RW | $\mathbf{75.10 \pm 0.10}$ | $\mathbf{0.003}$ |
| LAUREL-RW+LR | $\mathbf{75.20 \pm 0.07}$ | $\mathbf{1.68}$ |
| LAUREL-RW+LR+PA | $\mathbf{75.25 \pm 0.09}$ | $\mathbf{2.40}$ |

increase the model's accuracy by $0.25\%$ to reach $75.20\%$, while adding $4.37\%$ new parameters. With that in context, applying LAUREL on the model improves it (Table 1).

If we only use the LAUREL-RW version, we get an improvement of $0.15\%$ on average with only $0.003\%$ extra parameters, which is essentially negligible. When we try the LAUREL-RW+LR version from (5) with $r = 16$, we achieve $75.20\%$ accuracy while adding only $1.68\%$ new parameters; this matches the performance of the baseline with an extra layer, while using $2.6\times$ fewer extra parameters. Additionally, when we use the combined LAUREL-RW+LR+PA version from (6), we improve the accuracy to $75.25\%$ while still using $1.82\times$ fewer extra parameters than the baseline with one extra layer, demonstrating that LAUREL is superior to naively scaling the model. Notably, despite substantial changes to the residual connection, we did not find any training instabilities with LAUREL.

## 3.2. Large-Scale LLM Pre-training

In this setup, our goal was to test the performance of LAUREL when applied on top of strong LLMs. During the course of our work we evaluated LAUREL on two separate LLMs, which we pre-trained from scratch. The first LLM (LLM-1B) is a 1B parameter model pre-trained with a data-mixture consisting of only text tokenss. The second LLM (LLM-4B) is a 4B parameter model that was pre-trained with a multi-modal and multi-lingual data mixture. Both LLMs were trained with $\sim 0.5T$ tokens.

What we varied across the two LLMs was to allow for different budgets for increasing model footprint (parameters, latency, memory, etc.) when applying LAUREL. For LLM-1B we allow a very small increase in these metrics ($\sim 0.01\%$ extra parameters, and nearly no latency increase). For LLM-

4B we allow a lenient, yet modest increase in parameters ($\sim 0.1\%$ extra parameters), and latency (1% increase).

Given the scale of LLMs today, both the budgets would be considered negligible. For instance, a $0.1\%$ increase in parameters for a 4B model will only correspond to 4M more parameters. As demonstrated in the ResNet-50/ImageNet experiments (Section 3.1), LAUREL outperforms naive scaling; see Section 4.5 for a more detailed comparison.

Our objective behind testing LAUREL in these conditions was to demonstrate its efficacy and ensure that it scales well across different LLM setups in the wild.

### 3.2.1. LLM-1B: VERY LOW ADDITIONAL FOOTPRINT

For our first baseline, we chose a 1B parameter decoder-only transformer-based model. We pre-trained both the baseline, and our experiment with LAUREL, from scratch; we use the LAUREL-RW and LAUREL-LR versions (with $r = 4$). Both the models were trained using 256 Google Cloud TPU v5e chips for approximately two weeks each, using a pre-training mixture consisting of only text data that included webpages, books, code, and translations.

It is worth noting that the combined LAUREL-RW+LR variant adds only 0.012% more parameters as compared to the baseline model. Since we chose $r = 4$, the number of parameters added by LAUREL-LR is $8ND$ and the number of parameters added by LAUREL-RW is $2N$, for a total of $2N(4D+1)$ additional parameters. Typically $N \in [10, 100]$ and $D \in [500, 5000]$. For the sake of illustration, assuming $N = 20$ and $D = 1000$ and using LAUREL-RW+LR leads to $160,040$ extra parameters in a 1B parameter model. Thus, the number of new parameters is dwarfed by that of the original model. Furthermore, the additional latency introduced by LAUREL-RW+LR was within the noise range.

We evaluated both the pre-trained baseline and LAUREL models on a host of common LLM tasks such as Q&A, NLU, Math, Code, etc; see Table 2 for the results. The task type and individual tasks are listed in the first and second columns respectively, and a higher score is better for all the tasks. LAUREL outperforms the baselines on all tasks except on the MBPP dataset where it was neutral. To reiterate, these improvements were achieved with only 0.012% extra parameters and nearly no increase in latency.

### 3.2.2. LLM-4B: LOW ADDITIONAL FOOTPRINT

In this second setting, we experimented with a 4B parameter decoder-only model with a similar token budget, but trained on a multimodal and multilingual corpus of tokens.

To compensate for a $4\times$ larger model, and also the fact that the dataset and the evaluation tasks are harder, we allowing a bigger footprint budget for LAUREL; $\sim 0.1\%$ extra pa-

*Table 2.* Evaluation results on LLM-1B as described in Section 3.2.1; a 1B parameter decoder-only LLM pre-trained from scratch with (a) once with the baseline model architecture, and (b) once using LAuReL on top. We evaluated both the models on a number of common evaluation benchmarks (higher is better for all task type and task combinations listed below). LAuReL variant outperforms the baseline on all but one dataset while adding only 0.012% extra parameters. Results that provide $\geq 2\%$ relative improvement over the baseline are highlighted in green and bold.

| Task Type | Task | Baseline | LAuReL |
|---|---|---|---|
| Math | Math (Hendrycks et al., 2021b) | 3.54 | **3.70** **(+4.51%)** |
| | GSM8K-CoT (Cobbe et al., 2021) | 8.34 | **8.79** **(+5.39%)** |
| General Reasoning | MMLU (Hendrycks et al., 2021a) | 25.72 | **25.89** (+0.06%) |
| Q&A | BoolQ (Clark et al., 2019) | 58.07 | **65.66** **(+13.07%)** |
| | TyDi QA (GoldP) (Clark et al., 2020) | 67.98 | **72.58** **(+6.76%)** |
| Sentence Completion | HellaSwag (Zellers et al., 2019) | 64.84 | **65.06** (+0.03%) |
| Code | HumanEval (Chen et al., 2021) | 18.29 | **18.90** **(3.33%)** |
| | MBPP (Austin et al., 2021) | 27.00 | 27.00 |
| | GSM8K-PAL (Cobbe et al., 2021) | 10.31 | **11.37** **(10.28%)** |

*Table 3.* Evaluation results on LLM-4B as described in Section 3.2.2; a 4B parameter decoder-only LLM pre-trained from scratch with (a) once with the baseline model architecture, and (b) once using LAuReL on top. We evaluated both the models on a number of common evaluation benchmarks (higher is better for all task type and task combinations listed below). LAuReL variant outperforms the baseline on all but two tasks, while adding only $\sim$ 0.1% extra parameters. Results that provide statistically significant improvement over the baseline are highlighted in green and bold.

| Task Type | Task | Baseline | LAuReL |
|---|---|---|---|
| Math | Math (Hendrycks et al., 2021b) | 14.70 | **15.30** **(+4.08%)** |
| | MGSM (Shi et al., 2023) | 20.0 | **23.09** **(+15.45%)** |
| General Reasoning | MMLU (Hendrycks et al., 2021a) | 49.85 | **51.12** **(2.54%)** |
| Reading Comprehension | Belebele (Bandarkar et al., 2024) | 58.40 | **63.23** **(+8.27%)** |
| | BookQA (Mihaylov et al., 2018) | 50.36 | **60.46** **(+20.05%)** |
| Translation | WMT23 (Kocmi et al., 2023) | 68.32 | 68.24 (-0.11%) |
| Multimodal | MMMU (Yue et al., 2024) | 32.22 | **36.33** **(+12.75%)** |
| | Coco-Caption (Lin et al., 2014) | 95.69 | **99.15** **(+3.61%)** |
| | DocVQA (Mathew et al., 2021) | 68.28 | 68.34 (+0.08%) |
| | TextVQA (Singh et al., 2019) | 60.07 | **62.64** **(+4.27%)** |

rameters and $\sim 1\%$ extra latency. Note that this is still a negligible increase in model parameters and latency.

To match these budgets, we set $r = 64$, which provides a favorable trade-off of providing more capacity to the low-rank matrices as specified in the LAuReL-RW+LR formulation in (6), while also meeting the above-mentioned budgets.

In this setting, both the baseline and the LAuReL experiment were trained using 1024 Google Cloud TPU v4 chips for slightly more than two days each. See Table 3 for the evaluation results of both the baseline model and the model with LAuReL. The task type and individual tasks are listed in the first and second columns respectively, and a higher score is better for all the tasks. LLM-4B had a sophisticated suite of evaluation tasks, including math, general reasoning, reading comprehension, translation, and multimodal tasks. All of the listed evaluation tasks are used by leading LLMs to evaluate model quality, further solidifying our confidence in the model evaluation setup.

To start off, two of the LLM-4B evaluation tasks, Math, and MMLU were common with LLM-1B (refer to Table 2 for LLM-1B results). It can be seen that LLM-4B is much

stronger than LLM-1B on both the common evals; $4.16\times$ better on the Math task, and $1.93\times$ better on the MMLU task. We attribute this to LLM-4B being $4\times$ larger, and having a more sophisticated pre-training mixture. LAuReL using the LAuReL-RW+LR variant improves on both Math and MMLU, which is pleasantly surprising given how powerful is LLM-4B. For other tasks LAuReL improves significantly over the baseline as well, except for WMT23 and DocVQA, where it was neutral.

In terms of costs, the model adds $\sim 0.1\%$ more parameters. This is still a very reasonable since it means additional 4M parameters on top of a 4B parameter model. In terms of latency, we tested on both server-side (Google CloudTPU) for cloud serving, and a leading smartphone for on-device inference. In both the benchmarks, we measure nearly 1–2% increase in latency for prefill and generation. There was no human perceptible difference in terms of time-to-first-token.

We did not try the LAuReL-PA version for the above LLM experiments, as the LLM training was expensive. However, we expect the LAuReL-PA results from the ResNet experiments to also hold in this case as well.

## 3.3. LAUREL-LR: Rank vs Accuracy

We note that for the LAUREL-LR version on the ResNet-50/ImageNet combination, there is a pattern in terms of the best accuracy achieved with different values of $r$. In the combined LAUREL-RW+LR version, we experimented with different values of $r$, and computed the average of the best accuracy@1 achieved over 5 trials; see Figure 3. From Table 1, with the LAUREL-RW version alone we already achieve an average best accuracy@1 of 75.10%, therefore for the combined LAUREL-RW+LR version we would like to see the accuracy exceeding that.

We observe that when $r$ is small ($r \in \{4, 8\}$), there is not a significant improvement over the baseline LAUREL-RW experiment. This could be because a very small $r$ acts as an information bottleneck in the low-rank product in (3). As $r$ increases, the accuracy reaches the maximum for $r \in \{16, 32\}$; beyond this, the accuracy seems to drop though still higher than the LAUREL-RW baseline.

We believe this unimodal phenomenon could be due to the number of parameters added to the model (which increases linearly in $r$), since this would also require appropriate tuning of hyperparameters such as the learning rate as well as the regularization penalty.

Another possible cause could be how the low-rank matrices $A$ and $B$ are initialized. For $A \in \mathbb{R}^{D \times r}$, we used the Xavier initialization for the ResNet experiment and a column orthogonal initialization for the LLM experiments[2] while $B$ was always initialized to zero; this is similar to the scheme used in Hu et al. (2022). We found that initialization made a significant difference in the performance of the LAUREL-LR variant, and we posit that further work studying and improving the initialization scheme of the low-rank matrices could lead to better performance of the variant.

In terms of the tuning required, since $r \ll D$, and if $D = 512, 768, 1024, \ldots$ as in typical LLMs, this leaves a small range of discrete values for $r$ (unlike real-valued hyperparameters such as learning rate, weight decay, etc). In our experience $r \in \{32, 48, 64\}$ work well for LLMs.

## 4. Efficiency

Large models (LLMs and beyond) have been scaled both in the number of parameters as well as the number of tokens to achieve better model quality that scaling laws (Hoffmann et al., 2022) promise. However, this is directly in conflict with keeping training and inference costs reasonably low (we describe these costs shortly). These competing forces have resulted in the emergence of models such as Gemini Flash and Nano (Gemini-Team et al., 2024), DeepSeek (DeepSeek-A.I. et al., 2024), etc., which have attractive cost

---

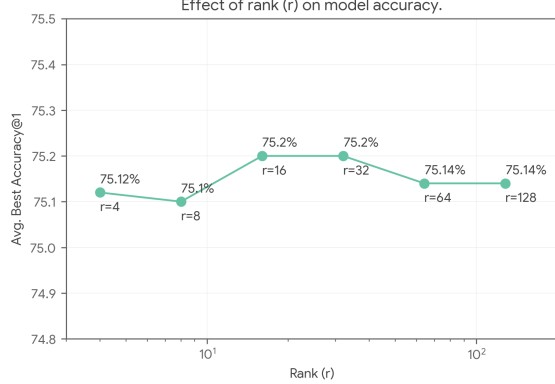

*Figure 3.* Average best accuracy@1 vs the rank ($r$) in the LAUREL-RW+LR variant.

versus quality trade-offs.

To achieve these favorable trade-offs, the above-mentioned models incorporate training techniques and architectural changes that can help improve quality while keeping training and inference costs low. Therefore, any proposed model efficiency technique including LAUREL should demonstrate not just improvement in quality, but also training and inference efficiency.

### 4.1. Efficiency Metrics

We now define some of the metrics on which we can measure model training and inference efficiency, before describing how LAUREL scores on them.

#### 4.1.1. NUMBER OF PARAMETERS

This is the most common metric designed to capture the cost of training and serving the model. A larger number of parameters implies larger forward and backward pass costs.

#### 4.1.2. LATENCY (TRAINING AND INFERENCE)

For training, a key metric to consider would be the number of steps taken per second. Similarly, another key metric would be the inference latency when deploying the model. For LLMs this can be broken down into the latency metrics in the warm-up stage (e.g., time-to-first-token, which can be further refined to prefill latency), and the generation stage (output tokens/second). For LLMs and other interactive models, both the time-to-first token and output tokens/second are important for a good user experience.

#### 4.1.3. PEAK MEMORY

Peak memory used during training is another key metric that is tracked to ensure that accelerators can accommodate the model graph and have enough free memory available

---

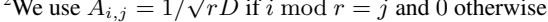

[2]We use $A_{i,j} = 1/\sqrt{rD}$ if $i \bmod r = j$ and 0 otherwise.

for forward and backward passes. Inefficiencies can be compensated by strategies like rematerialization (Kumar et al., 2019), which can help reduce peak memory usage but need recomputing some activations. Similarly, inference-time peak memory usage is also a key metric to track.

## 4.2. Efficiency Analysis

LAUREL variants are designed with the above efficiency metrics in mind, and in this section we study the performance of the variants on these metrics.

*Table 4.* Analysis of extra parameters, memory, and latency incurred for each LAUREL variant, per instantiation, except the LAUREL-PA case where at any time no more than $k$ previous activations are in-memory, incurring a total $\Theta(kD)$ extra memory cost. Also note that the latency cost of LAUREL-LR can be tighter than $O(rD^2)$, depending upon which matrix multiplication algorithm used. For simplicity, the latency bounds drop the batch and sequence dimensions.

| LAUREL VARIANT | PARAMS | MEMORY | LATENCY |
|---|---|---|---|
| LAUREL-RW | 1 OR 2 | $\Theta(1)$ | $O(1)$ |
| LAUREL-LR | $2rD$ | $\Theta(2rD)$ | $O(rD^2)$ |
| LAUREL-PA | $k$ | $\Theta(kD)$ | $O(kD)$ |
| LAUREL-LR+PA | $2rkD + k$ | $\Theta(2rkD + k)$ | $O(krD^2 + kD)$ |

Table 4 lists the costs associated for each LAUREL layer. We assume that we are working with a vector of size $D$ (the last dimension of the input). Note that the listed costs are *per*-LAUREL instantiation. Hence, if there are many transformer layers and if a particular LAUREL variant is used in each layer, then the relevant costs accumulate across the layers. We now examine the costs of the variants.

LAUREL-RW is the cheapest, with one or two scalars as parameters, a constant memory cost (the constant depending on the data type: `fp32`, `fp16`/`bf16`, `unit8`/`int8`, etc.), and a constant latency cost since the scalar multiplication might be optimized by the compiler.

The LAUREL-LR variant has $2rD$ parameters per instantiation ($rD$ for each of the two rank-$r$ matrices, $A$ and $B$), and uses $\Theta(rD)$ memory. The latency is upper bounded by $O(rD^2)$, since the input is a vector of dimension $D$ (ignoring the batch and sequence dimensions), and there are two matrix multiplication steps.

Finally, the LAUREL-PA variant uses $k$ previous activations. Assuming $h_i$ is the identity function, it uses $k$ extra parameters, and uses $\Theta(kD)$ extra memory for the previous activations. Since it requires $k$ extra vector additions, this

*Table 5.* Performance of LAUREL variants on a small LLM pre-training task. LAUREL experiments are based on top of BASELINE-24. LAUREL variants achieve a better test loss than BASELINE-28, while having fewer parameters, requiring lesser memory, and being faster in average step time.

| VARIANT | PARAMS (M) | TEST LOSS | PEAK MEMORY (GB) | AVG. STEP TIME (SEC) |
|---|---|---|---|---|
| BASELINE-24 | 157.20 | 3.0159 | 11.65 | 0.095 |
| BASELINE-28 | 179.23 | 2.9963 | 13.23 | 0.105 |
| *LAUREL VARIANTS (24 LAYERS)* | | | | |
| LAUREL-RW | 157.20 | 2.9557 | 11.93 | 0.095 |
| LAUREL-LR | 158.40 | 2.9624 | 12.29 | 0.098 |
| LAUREL-PA | 157.22 | 2.9512 | 12.55 | 0.100 |
| LAUREL-RW+LR | 158.40 | 2.9531 | 12.57 | 0.099 |
| LAUREL-RW+LR+PA | 160.83 | 2.9499 | 12.90 | 0.104 |

introduces an additional latency of $O(kD)$.

## 4.3. Ablation Study of LAUREL Variants

To supplement the efficiency analysis in Section 4.2, we provide an ablation study of LAUREL variants on a small LLM pre-training baseline. We used the C4 corpus (Raffel et al., 2020) with $\sim 10$B tokens, and a $4 \times 4$ Google Cloud TPU v6e (Trillium) topology[3] for compute.

In order to simplify the comparison across many ablations (and also to avoid the noise in downstream evaluations at the 10B tokens scale), we report model performance using the test loss, a reasonable proxy for downstream model quality. For each model we report the number of parameters, test loss, peak memory reported by profiling tools, and the average step time. The last two metrics are proxies for the theoretical memory and latency bounds respectively, as mentioned in Section 4.2. Lower is better for all metrics.

We trained our main baseline with 24 layers (BASELINE-24) and 157.2M parameters, along with a larger baseline with 28 layers (BASELINE-28) and 179.2M parameters. We ran all the LAUREL variants and two combinations (RW+LR, RW+LR+PA) on top of BASELINE-24. For the LAUREL-LR variants and its combinations, we picked $r = 32$. Similarly for LAUREL-PA variant and its combinations, we chose $k = 3$. Table 5 shows the results.

As seen in the results, all LAUREL variants have a lower test loss than the BASELINE-24, while having a negligible impact on additional parameters, peak memory, or average step time. In fact, LAUREL variants perform better than even BASELINE-28 in terms of the test loss while using much fewer parameters, lower peak memory, and lower average step time.

---

[3] We expect similar results with a comparable GPU setup.

## 4.4. Practical Recommendations

Given the experiments and the analysis of individual variants and their combinations, LAURL-RW is clearly the first candidate to try. LAURL-RW+LR offers further improvements to model quality, and seems to provide the best trade-off in terms of quality improvements and the additional overhead. We validated this in the ResNet (Section 3.1) and LLM experiments (Sections 3.2.1, 3.2.2, 4.3).

The LAURL-RW+LR+PA variant also leads to improvements over LAURL-RW+LR. The main caveat is the additional $\Theta(kD)$ memory, which might need to be monitored. If it is a concern, it can be mitigated to some extent by choosing a small value of $k$ or by changing the model sharding/re-materialization schemes.

Given the above tradeoffs in loss, memory, step time, etc. we recommend trying the LAURL variants in the following order: RW $\rightarrow$ LR $\rightarrow$ RW+LR / PA $\rightarrow$ RW+LR+PA.

## 4.5. Comparison with Naive Model Scaling

One of the advantages of LAURL we would like to highlight is that it is competitive against naive scaling. Concretely, given a modest additional budget for parameters, latency, and memory, allocating that budget to LAURL variants is likely to produce larger gains than using that budget in naive scaling methods such as additional layers. We first observed this in the ResNet experiments (Section 3.1), where LAURL variants match the performance of naive scaling while adding 2.6× fewer parameters, thus showing the Pareto-efficiency of LAURL (versus naive scaling).

In the LLM experiments in Section 3.2, the gains achieved using LAURL variants are at the cost of $\sim$ 0.01–0.1% more parameters. A naive way to use this 'additional' budget would be to increase the model dimension ($D$) such that we match the extra parameter budget. However, this is may not be feasible due to hardware limitations and memory alignment issues. Another way to use this 'additional' budget would be to increase the vocabulary size. However, the gains would be limited since this will include more tokens only from the tail of the distribution and may not contribute much to the model quality. Interestingly, Karpathy (2023) reported that for the NanoGPT model (Karpathy, 2022) pre-training was sped up by $\sim$ 25% when the vocabulary size was made divisible by 64. Therefore naive scaling might also hurt metrics like latency and memory by making the model setup suboptimal on the hardware.

Additionally, we also conducted a detailed study comparing LAURL with naive scaling on the same LLM pre-training task as LLM-4B as mentioned in Section 3.2.2. The baseline model and the the LAURL experiments are similar to their counterparts in Section 3.2.2, except a configuration change, which allows adding a layer. Both have 40 layers

each, and are referred to as BASELINE-40 and LAURL-40 respectively.

The naive scaling baseline had 41 layers, and is referred to as BASELINE-41 henceforth. In Table 6, we present the respective number of parameters and average step times of the three models when training (forward + backward pass). A lower average step time is better. For BASELINE-41 and LAURL, we report the delta in number of parameters and average step time when compared to the original baseline.

Table 6. Comparison between BASELINE-40 (40 layers), BASELINE-41 (41 layers), and LAURL in terms of parameters and average step time (forward and backward pass).

| MODEL | PARAMS (B) | AVG. STEP TIME (SEC) |
|---|---|---|
| BASELINE-40 | 4.400 | 1.65 |
| BASELINE-41 | 4.560 (+3.63%) | 1.68 (+1.81%) |
| LAURL-40 | 4.404 (+0.1%) | 1.69 (+2.42%) |

Note LAURL-40 adds only +0.1% parameters and incurs a step time penalty of 2.42%. In terms of extra parameters added, LAURL-40 adds $\sim$ 36× fewer parameters than the extra parameters added by BASELINE-41. The latency of LAURL-40 is slightly higher than BASELINE-41 since each layer incurs minor additional overhead.

Table 7 shows the downstream quality of the baselines and LAURL-40 on 10 tasks across Math, General Reasoning, Reading Comprehension, Translation, and Mutimodal domains. LAURL-40 wins on all tasks, except WMT23 and DocVQA where it matches the baselines. It outperforms not just BASELINE-40, but also BASELINE-41, achieving between 2%–21% improvement over BASELINE-40's performance.

We posit that naive scaling by adding an extra layer or two in a very deep network does not always lead to a corresponding improvement in performance out of the box. For such networks, we might require hyperparameter tuning (learning rate, weight decay) to counter overfitting, or training on more tokens to realize the expected theoretical model performance predicted by the scaling laws.

It is also well known that with deeper networks, the residual stream starts to play a crucial role in model convergence and quality. We suggest that this is the primary reason why LAURL performs well. It augments the residual stream with learned components, allocating capacity for the network to learn the linear components of the input better.

*Table 7.* LAURELEL-40's comparison with naive scaling. Results that provide statistically significant improvement over the BASELINE-40 are highlighted in green and bold. The percentages in the last row of the LAURELEL-40 section indicate the relative improvement over the baseline.

| METHOD | MATH | | GENERAL REASONING | READING COMPREHENSION | | TRANSLATION | MULTIMODAL | | | |
|---|---|---|---|---|---|---|---|---|---|---|
| | MATH | MGSM | MMLU | BELEBELE | BOOKQA | WMT23 | MMMU | COCO-CAP | DOCVQA | TEXTVQA |
| BASELINE-40 | 14.20 | 20.29 | 48.83 | 57.92 | 47.11 | 67.72 | 33.77 | 97.29 | 66.87 | 60.86 |
| (4.40B PARAMS) | ± 0.88 | ± 3.16 | ± 0.81 | ± 3.42 | ± 4.06 | ± 0.20 | ± 3.11 | ± 4.41 | ± 2.67 | ± 2.86 |
| BASELINE-41 | 14.50 | 20.29 | 49.10 | 59.30 | 42.77 | 67.74 | 35.33 | 98.50 | 66.18 | 60.23 |
| (4.56B PARAMS) | ± 0.9 | ± 3.15 | ± 0.82 | ± 3.34 | ± 4.15 | ± 0.21 | ± 3.12 | ± 3.53 | ± 2.76 | ± 2.87 |
| LAURELEL-40 | **15.11** | **23.12** | **50.32** | **62.65** | **57.22** | 67.71 | **37.57** | **99.27** | 66.92 | **63.15** |
| (4.404B PARAMS) | ± 1.01 | ± 3.51 | ± 0.82 | ± 3.15 | ± 3.81 | ± 0.19 | ± 3.10 | ± 5.03 | ± 2.65 | ± 2.82 |
| | (+6.48%) | (+13.94%) | (+3.05%) | (+8.16%) | (+21.46%) | (-0.02%) | (+11.25%) | (+2.03%) | (+0.07%) | (+3.76%) |

## 5. Related Work

**Residual Stream.** DenseNet (Huang et al., 2017) connects every pair of layers in the network and hence in the basic variant of DenseNet, all the activations need to be in memory. This is prohibitively expensive for deep LLMs and other modern transformers. When introducing dense-blocks, all previous activations within the block need to be visible to any given layer within the block; this requires refactoring the model architecture into dense blocks.

On the other hand, LAURELEL requires minimal changes. In fact, in LAURELEL-PA, which is the most similar to DenseNet, we make three design choices to achieve memory efficiency and performance. First, each layer only looks at the $k$ past activations ($k = 3$ seems sufficient in our experiments). Second, we use low-rank linear functions to further reduce memory usage due to activations. Third, we use learned scalars ($\gamma_i, \gamma_{i-1}, \dots$) to weigh the previous activations (which we found to be crucial in practice), whereas DenseNet assumes a simple sum of the previous activations.

He et al. (2016) introduce variants of residual connections with different types of 'gating', which look similar to the LAURELEL-RW variant, except that they use a much larger number of parameters ($O(D^2)$ per layer), where LAURELEL-RW uses one or two extra parameters per layer. Highway Nets (Srivastava et al., 2015) is similar to LAURELEL-RW but they also use $D^2 + D$ parameters; furthermore, they incur additional latency due a full-rank matrix multiplication. Residual Gates (Savarese, 2016) also is similar to LAURELEL-RW, except they use ReLU as the gating function. However, LAURELEL is a more general formulation.

**Architectural Changes.** Our work is inspired by recent model architecture improvements such as LoRA (Hu et al., 2022) and AltUp (Baykal et al., 2023) amongst others. LoRA is designed to efficiently fine-tune large pre-trained

models and it works directly on the model weight matrices level by introducing low-rank 'adapter' weights that are learned during the fine-tuning stage, while other model weights are held constant. In contrast, LAURELEL works at the residual connection level, which likely spans multiple weight matrices involved in the function $f$; furthermore, it is applied during the pre-training stage.

AltUp (Baykal et al., 2023) is designed to replicate the quality improvements of a model with a large model dimension, without having to pay the additional cost. It operates at the transformer-block level, constructing parallel 'lightweight' transformer blocks to approximate the model dimension scaling effect. In contrast, LAURELEL does not aim to replicate the dimension scaling effect.

Interestingly, LAURELEL can be applied in conjunction with both LoRA (during fine-tuning) and AltUp (during pre-training and fine-tuning). LAURELEL can also be enabled at the same time as parameter-sharing techniques.

## 6. Conclusion

In this paper we introduce the LAURELEL framework, which is a novel architectural change and a generalization of the residual/skip connection aimed at improving the model quality without significantly increasing the model size or latency. We study three versions (LAURELEL-RW, LAURELEL-LR, LAURELEL-PA) that can be mixed-and-matched together.

Through experiments, we demonstrate the efficacy of replacing the conventional residual connection with LAURELEL on both vision and language tasks, while also providing evidence for its advantages over naive model scaling methods. For future work, we aim to further improve LAURELEL by developing new variants with better trade-offs between quality and model footprint.

## Acknowledgments

We thank Cenk Baykal, Erik Vee, Fotis Iliopoulos, Khoa Trinh, and Sushant Sachdeva (in alphabetical order) for many helpful discussions. We would also like to thank Andrew Tomkins for suggesting the name for the work.

## Impact Statement

This paper presents work whose goal is to advance the field of Machine Learning. There are many potential societal consequences of our work, none which we feel must be specifically highlighted here.

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
