# OpenReview forum: "LAuReL: Learned Augmented Residual Layer"
_ICML.cc/2025/Conference — ICML 2025 poster_

### Official Review · Reviewer_MkJ5 · 2025-02-27

**Overall Recommendation:** 4

**Summary:**

Authors propose extensions to ResNet blocks that can improve performance with minimal addition of parameters. The extensions modifies the residual connection by adding a learnable transformation to it and/or utilises the output of previous layers.

**Claims And Evidence:**

OK, see below.

**Essential References Not Discussed:**

There is a study from Kaiming He et al from 2016 (Identity Mappings in Deep Residual Networks) which also quite extensively considerers different versions of ResNet blocks. The authors should check it and cite in their paper.

Another study which might be relevant is Huang et al, Densely Connected Convolutional Networks, 2018. They also describe approach to use previous activations, but not in ResNet style.

**Experimental Designs Or Analyses:**

OK, see below.

**Methods And Evaluation Criteria:**

OK, see below.

**Other Comments Or Suggestions:**

The manuscript is well written. I didn't find any typos.

**Other Strengths And Weaknesses:**

The authors mainly consider augmented learned residual layers. Specifically they propose three variants: 1) residual weight, 2) low-rank approximation, and 3) one using previous activations.

About the first one (RW), I am not sure there is enough novelty. He et al 2016 proposes quite similar form, even thought they do not train the parameters in similar manner as in this Laurel version. Also, He et al 2016 proposes to use 1x1 convolution for the residual layer, which is especially useful if $f(x)$ part changed dimension (e.g. different number of input and output channels in convolutions). This can perhaps be thought to be extended version of this Laurel but of course is also much more expensive and not exactly same. On the other hand, I am not fully confident that these introducing these two learnable parameters are meaningful in all cases. I would guess that $\alpha$ and $\beta$ could be either be included to the weights in $f$ of this or next layer or would the effect would disappear if layer or batch normalisation is applied. Anyway, the results show some kind of improvement, but I am not sure if this depends on the version of ResNet block used (see He et al 2016 for different options). The authors could perhaps experiment with it more.
The second one (low rank LR) seems novel and interesting and there are clearly gains shown by results (albeit their results are for combination of the first and second, but considering above, I think this LR is the key factor).
The third one (Laurel PA) is again a question. They did not have experiments using PA version only, but only for the version which combines all of these version. It might be that all gains are coming from LR.

To summarize: as a whole, I think this study is interesting and results looks good. However, I would wish to see Hu et al 2016 to be cited and reflect their work against this study. Also I would like to see experiments with LR and PA alone (e.g. in ResNet case) to see which one contributes has the highest contribution.

**Questions For Authors:**

No questions.

**Relation To Broader Scientific Literature:**

OK, see Strengths and Weaknesses.

**Theoretical Claims:**

Does not apply.

---

> ### Author Rebuttal · Authors · 2025-04-01
>
> Thank you for your comments and suggestions.
>
> #### __Ablation study with practical footprint metrics__
> Thank you for suggesting this experiment. This section demonstrates improvements of LAuReL variants individually, as well as when they are combined.
>
> For the purpose of comparing different LAuReL variants on the LLM pre-training task, we set up the following baseline. We pre-trained on the C4 corpus with $\approx$ 10B tokens. We used a $4 \times 4$ Google Cloud TPU v6e  topology, but we expect similar results with a comparable GPU setup.
>
> In order to simplify the comparison across many ablations (and also to avoid the noise in downstream evals at 10B token scale), we report model performance using the test loss, which is a good proxy for downstream model quality.
>
> We train our main baseline with 24 layers and 157.2M params, along with a larger baseline with 28 layers for comparison. We run all the LAuReL variants (RW, LR, PA) on top of the regular baseline with 24 layers. We also run two combinations of variants (RW+LR, RW+LR+PA). We report the number of params, test loss, peak memory as reported by profiling tools, and average step time. Lower is better for all metrics. The table below shows the results; note that all LAuReL experiments use L=24.
>
> | Variant                 | Params(M)    | Test Loss  | Peak Mem(GB)  | Avg. Step(sec) |
> |-------------------------:|:---------:|:---------:|:-----------:|:--------------:|
> | Baseline (L=24)         | 157.20  | 3.0159    | 11.65     | 0.095         |
> | Baseline-Large (L=28)   | 179.23  | 2.9963    | 13.23     | 0.105         |
> | LAuReL-RW                | 157.20 | 2.9557    | 11.93     | 0.095         |
> | LAuReL-LR                 | 158.40  | 2.9624    | 12.29     | 0.098         |
> | LAuReL-PA                | 157.22  | 2.9512    | 12.55     | 0.100         |
> | LAuReL-RW+LR         | 158.40  | 2.9531    | 12.57     | 0.099         |
> | LAuReL-RW+LR+PA | 160.83  | 2.9499    | 12.90     | 0.104         |
>
> All LAuReL variants perform better than the large baseline in terms of the test loss while using much fewer parameters, lower peak memory, and lower average step time. Given the above tradeoffs in loss, memory, step time, etc. we recommend trying the LAuReL variants in this order: RW $\rightarrow$ LR $\rightarrow$ PA / RW+LR $\rightarrow$ RW+LR+PA.  We will add these numbers in the revision.
>
> #### __Can α and β be learned by the network__
> That’s a good question. Since $f(.)$ is non-linear, absorbing $\alpha$ within the function is not equivalent to having an explicit $\alpha$  outside $f(x)$. Additionally, the normalization would apply on top of the weighted combination of $f(x)$ and $x$, so the individual scalars would still be useful in learning the relative weights of the two components. We also demonstrate that the -RW variant is useful on its own in the above ablation study.
>
> #### __Relevant Work__
> Thank you for the references.  DenseNet (Huang et al., 2018) connects every pair of layers in the network and hence in the vanilla version, all the activations need to be in memory. This is prohibitively expensive for deep LLMs and other modern transformers. When introducing dense-blocks, all previous activations within the block need to be visible to any given layer within the block; this requires refactoring the model architecture into dense blocks.
>
> On the other hand, LAuReL requires minimal changes. In fact, in LAuReL-PA, which is the most similar to DenseNet, we make three design choices to achieve memory efficiency and performance. Firstly, each layer only looks at the past $k$ activations. For the above experiments, $k=3$ was sufficient. Secondly, we also propose using low-rank linear functions to further reduce memory usage due to activations. Thirdly, the LAuReL-PA variant uses learned scalars ($\gamma_{i}$, $\gamma_{i-1}$, …) to learn the weights of the previous activations (which we found to be crucial), whereas DenseNet assumes a simple sum of the previous activations.
>
> Additionally, as seen above, LAuReL-RW and -LR variants provide significant improvements over naive-scaling, and can be combined with the -PA method.
>
> Identity Mappings in Deep Residual Networks (He et al., 2016): This paper introduces variants of residual connections with different types of 'gating', which look similar the -RW variant, except that they use a much larger number of parameters for either the exclusive gating  or the 1x1 conv gating  ($D_in \times D_out$ params per layer), which is much more expensive than the 1/2 params per layer of the -RW variant. For the rest of the paper, the authors introduce and focus on the pre-activation residual connection which places the activation functions such that it helps with optimization.
>
> We will include these discussions in the revision.
>
> We hope these address your concerns - thanks!

---

### Official Review · Reviewer_qGJS · 2025-03-13

**Overall Recommendation:** 4

**Summary:**

The paper introduces Learned Augmented Residual Layer (LAUREL), a novel enhancement to residual connections in CNNs and Transformers. LAUREL enriches the residual stream by incorporating learned scalar parameters and low-rank transformations, improving efficiency and expressivity.

Key Contributions:
1. Three variants (LAUREL-RW, LAUREL-LR, LAUREL-PA) balancing expressivity and efficiency.
2. Improves performance in vision (ResNet-50, ImageNet-1K) and language models (1B, 4B LLMs) with minimal parameter, latency, and memory overhead.
3. Matches naïve scaling accuracy on ImageNet-1K while using 2.6× fewer parameters.
4.  Boosts reasoning, math, reading comprehension, translation, and multimodal tasks with negligible overhead.

**Claims And Evidence:**

The claims presented in the submission are strongly supported by experimental evidence provided. The key claims of superior performance and efficiency are substantiated through well-structured experiments comparing variants of LAUREL against standard baseline models and naïve scaling. Results across ResNet-50 and LLMs convincingly support the authors' assertions about improved model efficiency.

**Essential References Not Discussed:**

N/A

**Experimental Designs Or Analyses:**

The experimental design is sound and well-structured. The evaluation on ImageNet-1K and multiple well-established LLM benchmarks is methodologically rigorous. The authors used clear baselines, performed multiple trials to report statistical significance (mean and standard deviation), and conducted parameter sensitivity analyses. One minor area for deeper exploration could be a broader ablation on initialization schemes, but the current treatment is already thorough and sufficient for the current claims.

**Methods And Evaluation Criteria:**

The claims presented by the authors—namely, LAUREL’s efficiency in parameter utilization, computational latency, and memory overhead relative to naïve scaling—are convincingly supported by clear empirical evidence presented through systematic experiments. The authors conducted extensive experiments, comparing multiple LAUREL variants with standard baselines across tasks from image classification (ImageNet-1K) and various LLM evaluation benchmarks (MATH, GSM8K, BoolQ, MMLU, etc.), thus substantiating their claims effectively. No problematic claims were identified.

**Other Comments Or Suggestions:**

N/A

**Other Strengths And Weaknesses:**

Strengths:

1. The proposed method is conceptually straightforward yet impactful, providing clear theoretical motivation and intuitive interpretation.
2. Significant empirical evidence across multiple model architectures (ResNet and transformers) and domains (vision and NLP) convincingly demonstrates general applicability.
3. Clear, thorough, and insightful experimental analysis, including ablation studies and analysis of the trade-offs regarding rank (r) parameters.

Weaknesses:

1. The paper could provide deeper insights or theoretical justification into why specific variants perform better under certain conditions.
2. Practical considerations for very deep networks (e.g., scaling to 100B+ parameter models) could be discussed to better position the approach for current state-of-the-art LLM scale.

Overall, the strengths outweigh the minor weaknesses mentioned above.

**Questions For Authors:**

N/A

**Relation To Broader Scientific Literature:**

The paper is appropriately situated within the broader scientific literature on deep learning efficiency, clearly citing key related work such as LoRA, AltUp, transformer block simplifications, model compression techniques, and distillation methods. The authors articulate how LAUREL differs and complements these approaches, providing a clear perspective of novelty and positioning their contribution clearly as a generalization of residual connections with lightweight augmentations.

**Theoretical Claims:**

The paper does not contain any explicit theoretical proofs that require verification. Instead, the contributions are algorithmic and empirical. Mathematical formulations provided (equations defining LAUREL variants) appear correct and straightforward.

---

> ### Author Rebuttal · Authors · 2025-04-01
>
> Thank you for your kind comments and valuable feedback!
>
> #### __Justification into why specific variants perform better under certain conditions__
> Deep networks generally do better with reasoning, math, coding etc. tasks. However, residual connections are crucial for such networks, and LAuReL helps augment these residual connections with learned components. So at a high-level we expect LAuReL to help with reasoning/math/coding tasks.
>
> Specifically we have the following intuition for why the different variants work:
>
> - LAuReL-RW: Helps with learning the importance of the residual input ($x_i$). This might not be as useful in the earlier layers, but would be important in other later layers to tackle the vanishing gradient problem using a higher learned weight for the residual input.
>
> - LAuReL-LR: Helps with allocating learning capacity for the linear part of the network (the residual input, i.e., $x_i$ in Figure 2), such that the main network can use it’s capacity towards learning better nonlinear functions ($f(x)$), while LAuReL contributes the linear components ($x_i + ABx_i$) to the residual stream.
>
> - LAuReL-PA: A hybrid of -RW and -LR variants, where multiple previous activations are used in a learned weighted manner, along with learning linear functions on top of them. This allows layers accelerated access to previous activations, along with learning their relative importance.
>
> Overall, LAuReL provides a general formulation for the residual connection, along with practical variants that can operate on the residual stream in different ways as highlighted above. LAuReL variants can be combined with each other, as seen in the ResNet experiments, as well as the small-scale ablations on an LLM reported in response to Reviewer gnJr.
>
> Interestingly, LAuReL variants consistently show better performance than naively adding a layer on top of the baseline (ResNet experiments in Table 1, small-scale LLM ablations in response to Reviewer gnJr, and LLM-2 naive scaling experiments in response to Reviewer jdcS). This demonstrates that operating on the residual stream using LAuReL has a non-trivial impact on model convergence, which cannot be matched by adding a full additional layer whereas LAuReL variants add $\leq$ 0.1% params in these experiments.
>
> #### __Practical considerations for very deep networks (e.g., scaling to 100B+ parameter models)__
> While it is hard to experiment with 100B+ params, we can extrapolate from the ablations done in the paper as well as in response to Reviewer gnJr and Reviewer jdcS.
>
> - The -RW variant is straightforward to try and does not have a tangible impact on latency and memory. We expect initialization of these scalars to also be important. For earlier layers, the initial weight given to the non-linear component can be higher.
> - The -LR variant is cheap enough to also be included. We expect a slight increase in memory and latency footprint. $r$ should be scaled up as $D$ grows. In our experiments up to 4B params, the ideal ratio of $D / r$ was between 24--32.
> - The -PA variant is also helpful if the network is very deep. Using $k={3, 4, …}$ was helpful in smaller-scale LLM experiments. When keeping a larger value of $k$, the accelerator memory usage should be monitored and appropriate rematerialization strategies should be employed. Although we do not see a large increase in memory usage with the -PA variant in the smaller-scale experiments.
> If the above variants work well with some headroom available in latency and memory, a combination of RW+LR or RW+LR+PA can be tried.
>
> We hope these address your concerns - thanks!

---

### Official Review · Reviewer_jdcS · 2025-03-13

**Overall Recommendation:** 4

**Summary:**

The paper introduces a new method for residual connections with an additional layer called Laurel. Their method involves introducing learnable parameters into the residual stream, which the authors argue might be too restrictive in its original form. The learnable parameters allow the authors to decide how much information might be incorporated from different parts of the residual stream.

The authors introduce three methods for Laurel. Laurel-RW is the simplest version that only introduces two learnable weights. Laurel-LR introduces a low-rank approximation of the residual stream. Laurel-PA applies the same approach as LR except over activations from prior layers. The authors test variants that apply these methods in tandem. The authors show that their method allow for better scaling, where their methods adds fewer parameters but leads to a larger increase.

**Claims And Evidence:**

In general, authors report consistent improvements with Laurel in comparison to naive scaling of their base architectures. But, in general, I have the following concerns:

* In most settings, Laurel seems best with Laurel-RW+LR or Laurel-RW+LR+PA settings. In general, using the low-rank approach seems tricky to me because you now have a new hyperparameter to tune. The authors generally address this with Figure 3 -- but I remain concerned with the trend of selecting the best r. The Figure doesn't show consistent results on when the best r is actually applicable. In general, the results also seem sensitive to r if I read Figure 3 correctly.
* I found the description in Section 4.4 for LLMs to be vague. For instance, I think the argument with ResNets is convincing where the authors show that the addition of a single layer compared to Laurel saves 2.6x parameters. Could the authors provide the same numbers for language models? If I add an additional layer to language model i.e. their 1B/4B model, how many parameters am I adding? The authors should discuss depth/parameter scaling in LLMs as well. It would be useful to have these upper bound memory costs in the table if possible.
* Under what setting would I use Laurel-PA? It seems to add additional overhead for using sharding, etc.
* The results of Laurel in Table 2/3 is really confusing to me. See questions. Overall, could error bars be added to results to make significance more obvious? Beyond this, do authors have any intuition on why certain baselines improve significantly while others stay relatively similar.

**Essential References Not Discussed:**

The authors may want to reference the many modifications that have been introduced to residual connections that overlap with their work in the related work. I think a section referring to HighwayNets or ResidualGates. Many similar ideas have been proposed and Laurel should discuss and cite them. Space is available so I recommend an additional related work subsection.

**Experimental Designs Or Analyses:**

* If the authors' aim is to have a parameter efficient scaling technique, why not compare with techniques that allow for scaling like parameter sharing or the methods that save over transformer blocks? I'm curious to see how much these methods in save in comparison to Laurel.

**Methods And Evaluation Criteria:**

Yes; Authors use traditional computer vision and NLP benchmarks for their evaluation.

**Other Comments Or Suggestions:**

didn't find any typos

**Other Strengths And Weaknesses:**

N/A

**Questions For Authors:**

* What do green percentages mean in Table 2/3? I found this very confusing since the percentages didn't match improvements and some were highlighted green. This isn't explained in the text as far as I can tell although I may be missing something.

**Relation To Broader Scientific Literature:**

The authors compare against other parameter efficient scaling methods, highlighting that theirs is parameter efficient.

**Theoretical Claims:**

N/A

---

> ### Author Rebuttal · Authors · 2025-04-01
>
> Thank you for your comments and suggestions.
>
> #### __Tuning $r$__
> Since is $r$ the rank of $A$, $B^T$, we expect $r \ll D$. Indeed, if $D = 512, 768, 1024, …$, this leaves a small range of discrete values for $r$ (unlike hyperparameters such as learning rate, weight decay, that can take continuous values). In our experience $r = {32, 48, 64}$ work well for LLMs.
>
> #### __LLM Depth / Parameter Scaling with Error Bars:__
> Similar to the comparison with naive depth-scaling with ResNet (Sec 3.1), we ran scaling experiments with LLM-2. Originally, both baseline/LAuReL had 40 layers. Adding the 41st layer required turning off a minor architectural change (which enforced the number of layers to be divisible by 2). To be fair, we re-ran the baseline and LAuReL.
>
> We present the number of params and average training step times.
>
> | Model | Params(B)   | Avg. Step (sec)|
> |--------------:|:---------|:---------------|
> | Baseline (40 layers)      | 4.40    | 1.65          |
> | Baseline$^{+1}$  (41 layers)  | 4.56 (+3.63%)   | 1.68  (+1.81%)        |
> | LAuReL        | 4.44 (+0.1%)   | 1.69 (+2.42%)           |
>
> Note LAuReL adds only 0.1% parameters and incurs a step time penalty of 2.42%. On latency, it is slightly above Baseline$^{+1}$ . This is because LAuReL is invoked for each layer, but the per-layer invocation cost is small.
>
> The table below shows the downstream quality of the baselines and LAuReL.
>
> | Model                                                                    | Math             | MGSM             | MMLU             | Belebele         | BookQA           | WMT23            | MMMU             | Coco-Cap         | DocVQA           | TextVQA          |
> |:------------------------------------------------------------------------------------|:----------------:|:----------------:|:----------------:|:----------------:|:----------------:|:----------------:|:----------------:|:----------------:|:----------------:|:-----------------:|
> | Baseline (40 layers) 4.40B Params                                     | 14.20 ±0.88     | 20.29 ±3.16     | 48.83 ±0.81     | 57.92 ±3.42     | 47.11 ±4.06     | 67.72 ±0.20     | 33.77 ±3.11     | 97.29 ±4.41     | 66.87 ±2.67     | 60.86 ±2.86     |
> | Baseline$^{+1}$ (41 layers) 4.56B Params                                   | 14.50 ±0.9      | 20.29 ±3.15     | 49.10 ±0.82     | 59.30 ±3.34     | 42.77 ±4.15     | **67.74** ±0.21     | 35.33 ±3.12     | 98.50 ±3.53     | 66.18 ±2.76     | 60.23 ±2.87     |
> | Laurel (40 layers) 4.44B Params     | **15.11** ±1.01 | **23.12** ±3.51 | **50.32** ±0.82 | **62.65** ±3.15 | **57.22** ±3.81 | 67.71 ±0.19     | **37.57** ±3.10 | **99.27** ±5.03 | **66.92** ±2.65 | **63.15** ±2.82 |
> | _Laurel %Change wrt Baseline_                                                          | _(+6.48%)_       | _(+13.94%)_      | _(+3.05%)_       | _(+8.16%)_       | _(+21.46%)_      | _(-0.02%)_       | _(+11.25%)_      | _(+2.03%)_       | _(+0.07%)_       | _(+3.76%)_       |
>
> LAuReL wins on all tasks except WMT23; it is more parameter-efficient than naive-scaling.
>
> #### __When to use Laurel-PA?__
> LAuReL-PA works well for deep networks where there is a risk of vanishing gradients.  A suitable rematerialization and sharding,  given the extra activations in memory, would help; the new ablation results (see Reviewer gnJr) are positive. We expect LAuReL-PA to do better than works such as DenseNet (Huang et al. '16), where the number of activations in-memory is quadratic.
>
> #### __Comparison with Parameter Sharing__
> Techniques like param-sharing are complementary, since they can be applied in conjunction with LAuReL.
>
> #### __Comparison with Highway Nets/Residual Gates__
> Highway Nets (Srivastava et al., 2015) is similar to LAuReL-RW, except that the Transform Gate requires ($D^2$ (weight matrix) + $D$ (bias)) params, in addition to the latency incurred by a full-rank matmult.  But we use 1-2 scalars per Laurel-RW layer, with no significant latency impact. Similarly, the Residual Gates (Savarese et al., 2017) also is similar to LAuReL-RW, except they use ReLU as the gating function. However, LAuReL is more general including variants like -LR, -PA, which can be combined.
>
> #### __Green Percentages in Table 2 / 3__
> Apologies. Green and bold font indicated statistically significant improvements for ResNet and LLM-2. The percentages for a couple of tasks (TyDiQA, MGSM, etc.) had a typo, but the absolute values were correct.
>
> #### __Intuition for asymmetrical improvement in tasks__
> Deep networks generally do better with reasoning/math/coding. However, residual connections are crucial and LAuReL helps augment them with learned components. So we expect LAuReL to improve such tasks.
>
> However, for certain other tasks the network might be bottlenecked on the number of parameters in the MLP layers. Thus it is hard to pinpoint why a particular task improves more than others with LAuReL.
>
> We hope these address your concerns and you can reconsider the assessment/score - thanks!

---

> > ### Comment · Reviewer_jdcS · 2025-04-02
> >
> > Thank you for your detailed response and the results in comparison to DenseNet are quite exciting as well as additional results reported here. I believe this addresses most of my concerns and I would recommend these results are added to the paper in the main paper or some appendix sections.
> >
> > I just had one clarification to ask just to make sure I understand the results. When you report *Laurel %Change wrt Baseline*, what is this actually measuring? Is it a combination of accuracy and the number of parameters saved? I'm still a little confused where these numbers come from although I may have missed something in the paper.
> >
> > One other remaining question I have is about scale. Do the authors have any intuitions about how Laurel will operate at larger scales? Of course the assumption is that at larger model sizes, the savings will be larger since adding additional layers may be expensive. But I'm curious whether Laurel will have the same model improvements. Will Laurel have any benefit in scaling models as well?

---

> > > ### Author Response · Authors · 2025-04-03
> > >
> > > Thank you for taking the time to go through our response carefully - glad that it addressed most of your concerns!  As you suggest, we will add these new numbers in the revision (since we cannot update the pdf now).
> > >
> > > #### __New results__
> > > Apologies for the confusion caused by brevity; please allow us to clarify.
> > >
> > > The results reported above are from new pre-training runs that we started to verify the scalability of LAuReL with LLMs as requested. They are similar to the results from LLM-2 (Table 3, Section 3.2.2), except that we had to re-run the baseline and LAuReL runs (both with 40 layers). This was because starting a naively-scaled baseline with an additional layer (i.e., with 40 + 1 = 41 layers) violated an implementation assumption, which expected the number of layers to be even. Thus, we had to disable this implementation assumption, re-run the baseline (to allow for a fair comparison), the naively-scaled baseline, and LAuReL.
> > >
> > > Regarding the metrics, the last row (‘LAuReL % Change w.r.t. Baseline’) in Table 2 reports the percentage improvement / regression on a downstream task achieved by LAuReL when compared to the Baseline. For example, in the ‘BookQA’ task, the baseline model scores 47.11, whereas LAuReL scores 57.22. This is an improvement of +21.46\% (i.e., 100 * ((57.22 / 47.11) - 1) = 21.46%). Similarly, on the MGSM task, the baseline model scores 20.29 while the model with LAuReL scores 23.12. This is an improvement of +13.94\%. Sorry about the non-descriptive label; we will clarify this in the revision.
> > >
> > > In terms of parameters, as seen in Table 1 LAuReL has +0.1\% more parameters than the baseline, and +2.42\% more step-time latency (forward + backward pass) than the baseline, however not only does LAuReL outperform the baseline, it also does significantly better than the naively scaled baseline which had one extra layer (referred to as Baseline$^+1$ with 41 layers in Table 1 & 2) on almost all tasks except WMT23, while using a fraction of the params as the naively scaled baseline.
> > >
> > > #### __Laurel at larger scales__
> > > That’s a good question! As you pointed out we expect that with deeper networks naive scaling to be less useful, while the residual connection will become even more important. Therefore, we expect LAuReL to continue to perform an important role in augmenting the residual connection, and hence improving model quality.
> > >
> > > For instance, we expect the -RW variant to learn that the weight of the linear component ($x$) and the non-linear component ($f(x)$) should vary across the layers. The -LR variant to free up more capacity for learning richer nonlinear functions ($f(.)$), and the -PA variant to further help with the vanishing gradient problem. For the -LR variant, expect $r$ to be scaled sub-linearly as $D$ is scaled, since $r \ll D$.
> > >
> > > We hope that we satisfactorily addressed your remaining concerns, and you can reconsider your score / assessment. Thank you for your time!

---

### Official Review · Reviewer_gnJr · 2025-03-14

**Overall Recommendation:** 2

**Summary:**

The paper introduces LAUREL (Learned Augmented Residual Layer), a new generalization of residual connections that can replace standard skip connections in neural networks. LAUREL outperforms traditional residual connections in both model quality and efficiency across vision and language tasks. When tested on ImageNet-1K, LAUREL achieved the same improvements as adding an entire extra layer while using 2.6× fewer parameters. In large language model experiments with 1B and 4B parameter models, LAUREL improved performance on downstream tasks by 2.54% to 20.05% while adding only 0.012% and 0.1% additional parameters respectively.

**Claims And Evidence:**

To support the claim, the authors conduct experiments on ResNet-50 on ImageNet-1K, and LLMs.

**Essential References Not Discussed:**

This paper is related to DenseNet, but iit does not discuss DenseNet-like works.

**Experimental Designs Or Analyses:**

The experimental designs are reasonable, but why only conduct experiments of ResNet on ImageNet, why not use ViT? ViT is very popular in computer vision.

**Methods And Evaluation Criteria:**

The proposed methods make sense for the problem or application at hand. However, the authors only shows theoretical extra memory, and latency incurred for each LAUREL. Do the authors have practical numbers?

**Other Comments Or Suggestions:**

Plese see weaknesses.

**Other Strengths And Weaknesses:**

Strengths:
1. Figure 2 clears show the idea of the paper and it is eaay to follow.
2. Experiments on ResNet and LLMs demonstrate the effectiveness of the proposed paper.

Weaknesses:
1. Do not report the pratical memory usage and latency compared with other methods? Especially on GPUs.
2. Lack of detailed ablation study for the design choises. How does different design choises influence pratical memeory usage and latency?
3. Lack of discussion with previous related methods, like DenseNet. In related work, the authors discuss Architectural Changes, Compression Techniques and Learning Techniques. Acutally, they are not very related to this paper. It is necessay to discuss the realtionship between this work and other DenseNet-like works.

**Questions For Authors:**

Plese see weaknesses.

**Relation To Broader Scientific Literature:**

This paper is related to DenseNet, but iit does not discuss DenseNet-like works.

**Theoretical Claims:**

There is no theoretical claims in the paper.

---

> ### Author Rebuttal · Authors · 2025-04-01
>
> Thank you for your comments and suggestions.
>
> #### __Ablation study with practical footprint metrics__
>
> Thank you for suggesting this experiment.  For the purpose of comparing different LAuReL variants on the LLM pre-training task, we set up the following baseline. We pre-trained on the C4 corpus with $\approx$ 10B tokens. We used a $4 \times 4$ Google Cloud TPU v6e  topology, but we expect similar results with a comparable GPU setup.
>
> In order to simplify the comparison across many ablations (and also to avoid the noise in downstream evals at 10B token scale), we report model performance using the test loss, which is a good proxy for downstream model quality.
>
> We train our main baseline with 24 layers and 157.2M params, along with a larger baseline with 28 layers for comparison. We run all the LAuReL variants (RW, LR, PA) on top of the regular baseline with 24 layers. We also run two combinations of variants (RW+LR, RW+LR+PA). We report the number of params, test loss, peak memory as reported by profiling tools, and average step time. Lower is better for all metrics. The table below shows the results; note that all LAuReL experiments use L=24.
>
> | Variant                 | Params(M)    | Test Loss  | Peak Mem(GB)  | Avg. Step(sec) |
> |-------------------------:|:---------:|:---------:|:-----------:|:--------------:|
> | Baseline (L=24)         | 157.20  | 3.0159    | 11.65     | 0.095         |
> | Baseline-Large (L=28)   | 179.23  | 2.9963    | 13.23     | 0.105         |
> | LAuReL-RW                | 157.20 | 2.9557    | 11.93     | 0.095         |
> | LAuReL-LR                 | 158.40  | 2.9624    | 12.29     | 0.098         |
> | LAuReL-PA                | 157.22  | 2.9512    | 12.55     | 0.100         |
> | LAuReL-RW+LR         | 158.40  | 2.9531    | 12.57     | 0.099         |
> | LAuReL-RW+LR+PA | 160.83  | 2.9499    | 12.90     | 0.104         |
>
> All LAuReL variants perform better than the large baseline in terms of the test loss while using much fewer parameters, lower peak memory, and lower average step time. Given the above tradeoffs in loss, memory, step time, etc. we recommend trying the LAuReL variants in this order: RW $\rightarrow$ LR $\rightarrow$ PA / RW+LR $\rightarrow$ RW+LR+PA.  We will add these numbers in the revision.
>
> We found that r={32, 48} work well for LAuReL-LR, and k={3, 4} works well for LAuReL-PA. We went with r=32, and k=3 respectively.
>
> #### __Relationship to DenseNet__
> Thank you for the reference.  DenseNet connects every pair of layers in the network and hence in the vanilla version, all the activations need to be in memory. This is prohibitively expensive for deep LLMs and other modern transformers. When introducing dense-blocks, all previous activations within the block need to be visible to any given layer within the block; this requires refactoring the model architecture into dense blocks.
>
> On the other hand, LAuReL requires minimal changes. In fact, in LAuReL-PA, which is the most similar to DenseNet, we make three design choices to achieve memory efficiency and performance. Firstly, each layer only looks at the past $k$ activations. For the above experiments, $k=3$ was sufficient. Secondly, we also propose using low-rank linear functions to further reduce memory usage due to activations. Thirdly, the LAuReL-PA variant uses learned scalars ($\gamma_{i}$, $\gamma_{i-1}$, …) to learn the weights of the previous activations (which we found to be crucial), whereas DenseNet assumes a simple sum of the previous activations.
>
> Additionally, as seen above, LAuReL-RW and -LR variants provide significant improvements over naive-scaling, and can be combined with the -PA method.
>
> We will include these discussions in the revision.
>
> #### __ViT__
> Unfortunately these experiments are still in-progress at the time of the rebuttal deadline; we will add them as soon as the runs finish.  However, we expect LAuReL to provide improvements on top of ViT baselines, given LAuReL has shown improvements on ResNet as well as three LLM baselines (LLM-1, LLM-2, and the new small LLM baseline mentioned above). Note that the latter three are transformer-networks, very much like ViT.
>
> We hope we have addressed your concerns and questions and we hope you can reconsider your assessment/score.  Thank you.

---

### Decision · Program_Chairs · 2025-05-01

**Decision:**

Accept (poster)

**Comment:**

The paper proposes to use previous activations to learn an "augmented" residual at the current layer. The review ratings are strong (3 accepts, 1 weak reject) overall. The results are general and strong,  with small overhead and simple designs. The AC checked the remarks of the weak reject review and found the author's rebuttal to be sufficient in addressing the concerns, especially regarding related work and missing metrics. The AC encourages the authors to include these content into the revision, and recommends acceptance.